# Using noise to probe recurrent neural network structure and prune synapses

**Eli Moore**
Department of Mathematics
University of California, Davis
Davis, CA 95616
`elimoore@ucdavis.edu`

**Rishidev Chaudhuri**
Center for Neuroscience
Department of Mathematics
Department of Neurobiology, Physiology and Behavior
University of California, Davis
Davis, CA 95616
`rchaudhuri@ucdavis.edu`

## Abstract

Many networks in the brain are sparsely connected, and the brain eliminates synapses during development and learning. How could the brain decide which synapses to prune? In a recurrent network, determining the importance of a synapse between two neurons is a difficult computational problem, depending on the role that both neurons play and on all possible pathways of information flow between them. Noise is ubiquitous in neural systems, and often considered an irritant to be overcome. Here we suggest that noise could play a functional role in synaptic pruning, allowing the brain to probe network structure and determine which synapses are redundant. We construct a simple, local, unsupervised plasticity rule that either strengthens or prunes synapses using only synaptic weight and the noise-driven covariance of the neighboring neurons. For a subset of linear and rectified-linear networks, we prove that this rule preserves the spectrum of the original matrix and hence preserves network dynamics even when the fraction of pruned synapses asymptotically approaches 1. The plasticity rule is biologically-plausible and may suggest a new role for noise in neural computation.

## 1 Introduction

The brain eliminates synapses, dramatically during development but then across the lifespan [1–3]. The degree of synaptic pruning post-learning correlates with learning performance, suggesting an important functional role [4, 5]. Moreover, connection density is disrupted across a spectrum of diseases [6–9]. Determining how the brain finds and maintains sparse network structure is important to understand the brain's remarkable energy efficiency and replicate it in artificial neural networks as well as to understand changes in connection density with aging [10, 11] and in disease [6–9].

In a highly-recurrent network with multiple pathways of information flow, it is difficult to determine which synapses are redundant and can be safely pruned, and which are important and should be retained. For example, even if a synapse between two neurons is strong, if information can travel between the neurons by alternative pathways then the synapse is redundant and can be removed. A biologically-plausible pruning rule must determine this higher-order structure using information locally available at the synapse.

Neural systems seem noisy, at multiple levels: neural activity contains large background fluctuations, responses to the same stimulus can be quite variable, and synapses often fail to propagate a signal [12–15]. Here we show that noise could play a useful computational role in synaptic pruning. Specifically, the pattern of activity correlations in a noise-driven network reflects higher-order network structure in exactly the form needed for good synaptic pruning (as predicted by a theoretical argument). We

construct a local plasticity rule that either strengthens or prunes synapses with a probability given by the synaptic weight and the noise-driven covariance of the neighboring neurons. The plasticity rule is unsupervised and task-agnostic, seeking only to preserve existing network dynamics, whatever they are. Thus, it could act alongside learning or during separate pruning epochs (e.g., sleep), and does not restrict the learning rule in any way.

We prove that, for a class of undirected linear and rectified linear networks, the pruning rule preserves multiple useful properties of the original network (including the spectrum and resting-state variances), even when the fraction of removed synapses approaches 1. The theoretical results link neural network pruning and noise-driven dynamical systems to a powerful body of results in sampling-based graph sparsification [16–18] and to matrix concentration of measure tools [19–22].

## 2  Problem setup

We primarily consider linear neural networks of the form

$$\frac{d\boldsymbol{x}}{dt} = -D\boldsymbol{x} + W\boldsymbol{x} + \boldsymbol{b}(t) = Ax + \boldsymbol{b}(t) \tag{1}$$

The vector $\boldsymbol{x}$ represents the firing rate of $N$ neurons, with $x_i$ the firing rate of the $i$-th neuron. $\boldsymbol{b}(t)$ is the external input to the neurons (including biases). $W$ is the matrix of weighted connections between the neurons, with $W_{ij}$ the connection strength from the $j$-th to the $i$-th neuron. $D$ is a diagonal matrix representing the intrinsic leak of activity (or the excitability of the neuron). Finally we define the matrix $A = -D + W$. We discuss generalizations to rectified linear networks in Section 6.

The pruning rule seeks to generate a sparse network with corresponding matrix $A^{sparse}$ with two properties. First, the number of edges in the network (i.e., number of non-zero entries in $A^{sparse}$) should be small when compared to the $\sim N^2$ possible edges in the original network. Second, the dynamics of the pruned network

$$\frac{d\boldsymbol{x}}{dt} = A^{sparse}\boldsymbol{x} + \boldsymbol{b}(t) \tag{2}$$

should be similar to the dynamics of the original network in Eq. 1.

To measure the similarity of $A$ and $A^{sparse}$, we adopt the notion of spectral similarity [16, 17] from the field of graph sparsification and require that for some small $\epsilon > 0$,

$$|\boldsymbol{x}^T(A^{sparse} - A)\boldsymbol{x}| \leq \epsilon|\boldsymbol{x}^T A\boldsymbol{x}| \quad \forall \boldsymbol{x} \in \mathbb{R}^N. \tag{3}$$

This notion of similarity is quite strong. For symmetric matrices it requires that the eigenvalues of $A^{sparse}$ approximate the eigenvalues of $A$ (and hence all the timescales of the resulting dynamics) to within a multiplicative factor $\epsilon$ (see SI S1.3). This closeness is much stronger than low rank approximation, which preserves only the largest eigenvalues. The pruning rule also approximately preserves matrix-vector products and eigenvectors corresponding to separated eigenvalues. The timescales and activity patterns of the dynamical system in Eq. 1 are determined by the spectrum of $A$, and thus spectrum-preserving sparsification will (approximately) preserve dynamics.

## 3  An unsupervised noise-driven anti-Hebbian pruning rule

Consider the network in Eq. 1 when driven by independent noise at each node. We set $\boldsymbol{b}(t) = \boldsymbol{b} + \sigma\boldsymbol{\xi}(t)$ where $\boldsymbol{b}$ is an arbitrary vector of constant background input to the network, $\boldsymbol{\xi}$ is a vector of IID unit variance Gaussian white noise, and $\sigma$ is the standard deviation of the input noise. Let $C$ be the covariance matrix of the firing rates in response to this white noise input. For the synapse from neuron $j$ to neuron $i$, with weight $w_{ij}$, define the probability

$$p_{ij} = \begin{cases} Kw_{ij}\left(C_{ii} + C_{jj} - 2C_{ij}\right) & \text{for } w_{ij} > 0 \quad \text{(excitatory)} \\ K|w_{ij}|\left(C_{ii} + C_{jj} + 2C_{ij}\right) & \text{for } w_{ij} < 0 \quad \text{(inhibitory)}. \end{cases} \tag{4}$$

Here $C_{ii}$ and $C_{jj}$ are the variances of the $i$th and $j$th neurons, and $C_{ij}$ is their covariance. $K$ is a proportionality constant and determines the density of the pruned network, which will have $NK/2$ total connections on average and thus average degree of $K/2$ per neuron (for unit variance noise and symmetric networks).

Now consider a pruning process that independently preserves each edge with probability $p_{ij}$ yielding $A^{sparse}$, where for $i \neq j$,

$$A_{ij}^{sparse} = \begin{cases} A_{ij}/p_{ij} & \text{with probability } p_{ij} \\ 0 & \text{otherwise.} \end{cases} \tag{5}$$

For the diagonal terms (i.e., leak / excitability) $A_{ii}^{sparse}$ we either preserve the original diagonal and set $A_{ii}^{sparse} = A_{ii}$ or define the perturbation $\Delta_i = \sum_{j \neq i} |A_{ij}^{sparse}| - \sum_{j \neq i} |A_{ij}|$ to be the change in total input to neuron $i$ and set $A_{ii}^{sparse} = A_{ii} - \Delta_i$. $\Delta_i$ is small with zero mean, and biologically corresponds to changing the excitability of neuron $i$ in response to a change in total input (excitability is known to be homeostatically regulated [23]). We call these the "original diagonal" and "matched diagonal" settings respectively. The proofs apply to the "matched diagonal" setting (empirically, similar results apply to the "original diagonal" setting). We will also refer to the pruning rule defined by Eqs. 4, 5 (in both settings for the diagonal) as **noise-prune** going forward.

The noise-prune rule is predicted by a theoretical argument (see next section), but has an appealingly simple interpretation and we here provide some intuition for why it might work. First, note that the probability to preserve a synapse depends on the magnitude of its weight, $|w_{ij}|$. Thus, all else being equal, synapses with larger weight are more important and are preserved. The remainder of the expression for the preservation probability is $(C_{ii} + C_{jj} \pm 2C_{ij})$, which we call the diff-cov term. This term can be slightly rewritten as $2\tilde{C}_{ij}(1 \pm C_{ij}/\tilde{C}_{ij})$, where $\tilde{C}_{ij} = (C_{ii} + C_{jj})/2$ is the mean variance of nodes $i$ and $j$. Thus, preservation probability is proportional to $\tilde{C}_{ij}$, reflecting that nodes with higher variance are considered more important and thus their connections are likely to be preserved (as in a PCA-like approximation). Finally, there is an anti-Hebbian term that for excitatory synapses takes the form $(1 - C_{ij}/\tilde{C}_{ij})$ (if $2\tilde{C}_{ij}$ is factored out) or $(C_{ii} + C_{jj} - 2C_{ij})$. Synapses are thus likely to be preserved if they are weakly or anti-correlated despite having an excitatory connection. The equivalent term for inhibitory synapses is $(C_{ii} + C_{jj} + 2C_{ij})$. The sign of the covariance is flipped, reflecting that inhibitory connections are expected to anti-correlate neurons.

The covariance of neurons $i$ and $j$ depends both on the strength of the direct connection between them (i.e., $w_{ij}$) and on indirect connections through the rest of the network. Neurons that are highly correlated are likely to have multiple indirect connections, suggesting that the direct connection is redundant and can be pruned (see schematics in Fig. 1a,b). More generally, the pruning rule can be understood as probing whether neurons are more correlated than expected given the weight of their direct connection. If they are, the connection is likely to be redundant.

For a simple example of why higher correlations indicate a connection that can be pruned, consider probing one's indirect connections to a friend by spreading a rumor about them and measuring how distorted the rumor is by the time they hear it (or, alternatively, playing the children's game "telephone"). More distortion (i.e., lower correlation) indicates that there are few indirect routes for information to flow through and thus the connection is likely to be important. Conversely, the less distorted the message, the more redundant the connection.

## 4   Proofs

We derive this pruning rule from a two-part theoretical argument. First, we consider a sampling-based approach to pruning that independently strengthens or removes each edge of a network with some probability (as in Eq. 5) and derive sampling probabilities that preserve network dynamics. The structure of the argument follows Spielman & Srivastava (2011), with slight extension to signed symmetric diagonally-dominant neural network matrices. Second, we show that these theoretically-derived probabilities have a surprisingly simple expression in terms of the covariance of the network activity when driven by noisy fluctuations. Thus, there exists a simple and biologically-plausible way for neural networks to compute the sampling probabilities using local information.

Note that the proof, but not the noise-prune rule itself, requires the matrix $A$ to be symmetric (corresponding to an undirected graph) and diagonally-dominant (corresponding to quite leaky neurons). These are strong restrictions that do not typically apply to neural networks, and we discuss generalizations and limitations later, including preliminary empirical results that show that noise-prune can work well even when these restrictions do not hold (see Fig. 2).

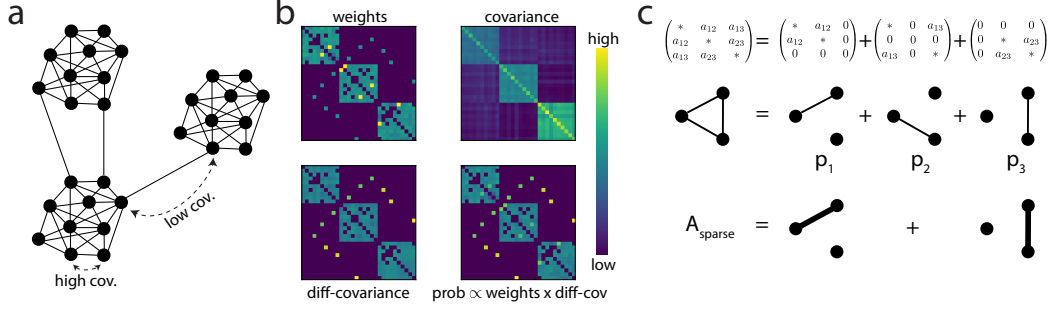

Figure 1: **A noise-driven unsupervised synaptic pruning rule.** (a) Schematic of a network where noisy fluctuations reflect higher-order connectivity structure. Network has 3 densely-connected clusters, with a few long-range connections. Covariance between neurons within a cluster is high compared to neurons participating in different clusters. (b) Pruning rule uses weights and covariances to identify important synapses. Top left: connection weights in a network with 3 densely-connected clusters and sparse connections between clusters. Also note the presence of a few strong connections within each cluster. Top right: covariance when driven by noise. Bottom left: difference of covariances (as in Eq. 4). Bottom right: sampling probabilities from pruning rule. The rule correctly identifies that the sparse connections between clusters are important and assigns them higher probability, along with the handful of exceptionally strong connections within a cluster. Most connections within a cluster are redundant and given lower probability. (c) Schematic of proof strategy. The original network (shown as a matrix in the top row and as a graph in the middle row) can be written as a sum of edge pieces. The edges are assigned sampling probabilities ($p_1$, $p_2$, $p_3$) that depend on weight and covariance. A given application of noise-prune yields a sparse network (bottom row) where some connections are preserved and strengthened (first and third edges) and others are pruned (second edge). For appropriate probabilities, the spectrum of $A^{sparse}$ is close to that of the original network.

## 4.1 Derivation of probabilities

Assume that the matrix $A$ from Eq. 1 is symmetric ($A_{ij} = A_{ji}$) and diagonally-dominant, meaning that $|A_{ii}| \geq \sum_{j \neq i} |A_{ij}| = \sum_{j \neq i} |w_{ij}|$. The diagonal entries of $A$ are negative, reflecting the leak, and thus $A$ is negative definite (note that eigenvalues must be negative for the linear system to be stable, but the argument can be extended to non-invertible matrices by working in the subspace orthogonal to the null space [16]). For notational convenience, we define the positive definite matrix $B = -A$ and consider $B$ instead of $A$ in this section.

Given an edge $(i, j)$, $i > j$, with weight $w_{ij}$, define the edge matrix $X^{(i,j)}$ to have $i$th and $j$th diagonal entries $X_{ii}^{(i,j)} = X_{jj}^{(i,j)} = |w_{ij}|$. Set the $(i,j)$th and $(j,i)$th off-diagonal entries $X_{ij}^{(i,j)} = X_{ji}^{(i,j)} = -w_{ij}$ and remaining entries 0 (Fig. 1c for a schematic). Thus, $X^{(i,j)}$ has off-diagonal pieces equal to negative edge weight and diagonal pieces equal to its magnitude. Also define $X_{ii}^{(i,i)} = B_{ii} - \sum_{j \neq i} |w_{ij}|$, with remaining entries 0. Because $B$ is diagonally-dominant with positive diagonal, the single non-zero entry of $X^{(i,i)}$ is positive. For simplicity, we consider matrices where $X^{(i,i)} = 0$, but it is straightforward to include non-zero $X^{(i,i)}$ (SI S1.1). $B$ can be written as a sum over edge matrices as $B = \sum_{i>j} X^{(i,j)}$.

Now define the random matrix $\tilde{X}^{ij}$ as

$$\tilde{X}^{ij} = \begin{cases} X^{(i,j)}/p_{ij} & \text{with probability } p_{ij} \\ 0 & \text{otherwise} \end{cases} \qquad (6)$$

And define $B^{sparse} = \sum_{i>j} \tilde{X}^{ij}$. For any choice of $p_{ij}$, $\mathbb{E}[B^{sparse}] = B$. Thus on average $B^{sparse}$ is the original matrix. Also note that the average number of edges in $B^{sparse}$, $\mathbb{E}[N_{edges}] = \sum_{i>j} p_{ij}$.

If the $p_{ij}$ are close to 1, then most edges will be included in any realization of $B^{sparse}$ and it will be close to $B$, but not sparse. If the $p_{ij}$ are small, then $B^{sparse}$ will be sparse but might be a poor approximation to $B$. A good algorithm will choose the $p_{ij}$'s to ensure both that $B^{sparse}$ is close to $B$ (in some appropriate sense) and that the number of non-zero edges is small.

To determine good sampling probabilities, we follow Spielman & Srivastava (2011) and first transform $B$ to the identity matrix. Note that $I = B^{-1/2}BB^{-1/2}$, where $I$ is the identity matrix and $B^{-1/2}$ is the matrix that squares to $B^{-1}$ (well-defined because $B$ is symmetric positive definite). Define $\tilde{Y}^{ij} = B^{-1/2}\tilde{X}^{ij}B^{-1/2}$ and $\tilde{I} = \sum_{i>j}\tilde{Y}^{ij} = B^{-1/2}B^{sparse}B^{-1/2}$. Note that $\mathbb{E}[\tilde{I}] = I$.

The matrix Chernoff bound [19–22] bounds the probability that $\tilde{I}$ is far from $I$. Let $M$ be an upper bound on the $\tilde{Y}^{(i,j)}$'s, so that $0 \leq ||\tilde{Y}^{(i,j)}||_2 \leq M$. Let $\tilde{\lambda}_{min}$ and $\tilde{\lambda}_{max}$ be the minimum and maximum eigenvalues of $\tilde{I}$. For given $0 < \epsilon < 1$, the bound guarantees that

$$P\left[\tilde{\lambda}_{min} \leq (1-\epsilon)\right] \leq N\left(e^{-\epsilon^2/2}\right)^{1/M} \text{ and } P\left[\tilde{\lambda}_{max} \geq (1+\epsilon)\right] \leq N\left(e^{-\epsilon^2/3}\right)^{1/M} \quad (7)$$

A good approximation thus requires that $M$ be small. On the other hand, since the sampled pieces are rescaled by $1/p_{ij}$, a sparser approximation (smaller $p_{ij}$) corresponds to larger $M$.

For each $(i, j)$, the maximum value that $||\tilde{Y}^{(i,j)}||_2$ takes is $\frac{1}{p_{ij}}||B^{-1/2}X^{(i,j)}B^{-1/2}||$. Set

$$\frac{p_{ij}}{K_{deg}} = ||B^{-1/2}X^{(i,j)}B^{-1/2}|| = \text{tr}(B^{-1}X^{(i,j)}) = |w_{ij}|(B_{ii}^{-1} + B_{jj}^{-1} - \text{sign}(w_{ij})2B_{ij}^{-1}), \quad (8)$$

for some constant $K_{deg}$, where the second equality holds since the trace is cyclic and equal to the 2-norm of a rank-1 positive semi-definite matrix. This equalizes the maximum value across $\tilde{Y}^{(i,j)}$, yielding $M = 1/K_{deg}$.

For any given $\epsilon$, ensuring that the probabilities in Eq. 7 are small requires that $K_{deg} \geq 4\log(N)/\epsilon^2$ (the constant 4 is chosen semi-arbitrarily to ensure small probability for reasonable $N$ and other values $> 3$ can be chosen). Thus $K_{deg} = 4\log(N)/\epsilon^2$ guarantees that the eigenvalues of $\tilde{I}$ lie within $[1-\epsilon, 1+\epsilon]$ with high probability (w.h.p.). Consequently, w.h.p., we have

$$(1-\epsilon)\boldsymbol{y}^T\boldsymbol{y} \leq \boldsymbol{y}^T\tilde{I}\boldsymbol{y} \leq (1+\epsilon)\boldsymbol{y}^T\boldsymbol{y} \quad \forall \boldsymbol{y} \in \mathbb{R}^N. \quad (9)$$

Given some $\boldsymbol{x} \in \mathbb{R}^N$, set $\boldsymbol{y} = B^{1/2}\boldsymbol{x}$ yielding that w.h.p.,

$$(1-\epsilon)\boldsymbol{x}^T B \boldsymbol{x} \leq \boldsymbol{x}^T B^{sparse}\boldsymbol{x} \leq (1+\epsilon)\boldsymbol{x}^T B \boldsymbol{x} \quad \forall \boldsymbol{x} \in \mathbb{R}^N. \quad (10)$$

And observing that $B = -A$ yields the desired approximation.

The average number of edges in the pruned network $\langle N_{edges}\rangle = \sum_{i>j} p_{ij}$ (and a standard scalar Chernoff bound shows that fluctuations around the mean are small). Note that $\sum_{i>j}||B^{-1/2}X^{(i,j)}B^{-1/2}|| = N$ (proof in SI S1.1). Hence $\langle N_{edges}\rangle = \sum_{i>j} p_{ij} = NK_{deg}$. Consequently, if $K_{deg} = 4\log(N)/\epsilon^2$ then, in terms of $\epsilon$, $\langle N_{edges}\rangle = 4N\log(N)/\epsilon^2$.

As with the sparsification of graph Laplacians [16], for a fixed relative approximation ($\epsilon$) to $A$, the number of edges in $A^{sparse}$ need only be $O(N\log(N))$. This is very strong: if the original network is dense then it has $\sim N^2$ edges; thus the fraction of edges needed for fixed $\epsilon$ goes to 0 with increasing $N$. On the other hand, if the number of edges in $A^{sparse}$ is a small but non-vanishing fraction of the edges in $A$, then the approximation becomes arbitrarily good with increasing $N$ (i.e., $\epsilon \to 0$).

## 4.2 Probabilities from noise-driven covariance

Consider the network of Eq. 1 when driven by uncorrelated white noise of variance $\sigma^2$ at each node. Set the constant background input $\boldsymbol{b} = 0$ for simplicity (this just shifts the mean to 0). The covariance matrix, $C$ of the resulting dynamics is $C = \mathbb{E}[\boldsymbol{x}\boldsymbol{x}^T]$ and satisfies the Lyapunov equation [24, 25]:

$$AC + CA^* = -\sigma^2 I. \quad (11)$$

Let $A$ be a normal matrix (meaning $AA^* = A^*A$, where $A^*$ is the conjugate transpose of $A$; this category includes symmetric matrices, such as the ones we consider). Define $A_{symm} = (A + A^*)/2$. It is straightforward to show that $C \propto A_{symm}^{-1}$ (see SI S1.2 for details). In particular, if $A$ is symmetric then $C = -\sigma^2 A^{-1}/2$. Substituting $2C/\sigma^2$ for $B^{-1} = -A^{-1}$ in Eq. 8 yields

$$p_{ij} = K|w_{ij}|(C_{ii} + C_{jj} - \text{sign}(w_{ij})2C_{ij}), \quad (12)$$

with $K = 2K_{deg}/\sigma^2$. Thus, perhaps surprisingly, the pattern of noise-driven correlations exactly encodes the optimal sampling probabilities predicted by the matrix Chernoff bound.

# 5   Numerical results

In Fig. 2 we show the performance of noise-prune (in the matched diagonal regime) on diagonally-dominant networks with clustered structure (parameters in figure caption). We compare it to a control case in which edges are sampled and either strengthened or pruned (as in Eq. 5) but with probabilities just proportional to weight (i.e., without a covariance term and thus without accounting for higher-order network structure). The proportionality constant for the control is chosen to match the expected number of edges preserved by noise-prune.

The box plots in the first columns of Fig. 2a,b show the distribution of relative change in eigenvalues of the pruned network when compared to the original network, given by $\epsilon_{\lambda_i} = \left| \frac{\tilde{\lambda}_i}{\lambda_i} - 1 \right|$, where $\tilde{\lambda}_i$ is the $i$th eigenvalue of $A^{sparse}$, and $\lambda_i$ is the $i$th eigenvalue of $A$. The box plots in the second column compare the relative change in quadratic forms $\epsilon_{v_i} = \left| \frac{v_i^T A^{sparse} v_i}{v_i^T A v_i} - 1 \right| = \left| \frac{v_i^T A^{sparse} v_i}{\lambda_i} - 1 \right|$ for the two approximations, where $(v_i, \lambda_i)$ is the $i$th eigenvector-eigenvalue pair of $A$. Lastly, the box plots in the third column measure how close the eigenvectors of the original network are to being eigenvectors of the pruned network using the normalized dot products of the eigenvectors before and after applying $A^{sparse}$: $\cos(\theta_i) = \frac{|v_i^T A^{sparse} v_i|}{\|A^{sparse} v_i\|}$. In all cases, noise-prune performs better than the control, with the performance improving as the networks get larger (panel a vs. b).

We also compare the dynamical response of networks to various inputs before and after pruning. In Fig. 2c we show the response of symmetric clustered networks to random inputs before and after pruning, and find that noise-prune preserves both the responses of individual nodes (left panel) and the network response trajectory as a whole (right panel). We also find similar preservation for structured inputs directed along the slow eigenvectors of the network coupling matrix, which reflect integrative shared dynamical modes that may be used for computation, Fig. 2d. Moreover, noise-prune significantly outperforms the purely weight-based strategy (red vs. blue) and thus using the higher-order structure reflected in the noise covariances dramatically improves the preservation of dynamics in the pruned network.

The theoretical results apply to the case of symmetric matrices but the pruning rule itself is quite general. We thus empirically characterize noise-prune on non-symmetric clustered networks for both random and eigenvector inputs, Fig. 2e and f. Again, noise-prune preserves network dynamics and does much better than a control strategy that relies only on weight, suggesting that good performance extends beyond the theoretical guarantees.

# 6   Extensions

We next briefly describe some extensions of the framework described above (further details in SI).

**Approximate probabilities**   The pruning is robust to approximate probabilities (as with graph Laplacian sparsification [16]). To see robustness, note that the probabilities (a) determine the upper bound $M$ used in Eq. 7 and (b) determine $\langle N_{edges} \rangle$ through their sum. Consequently, if some edges are undersampled by a multiplicative factor $\alpha < 1$ (i.e., probabilities $\hat{p}_{ij} = \alpha p_{ij}$ where the $p_{ij}$'s are the probabilities in Eq. 12) then the bound $M$ will be inflated by a factor of $1/\alpha$ and Eq. 10 will still hold albeit with a larger $\hat{\epsilon} = \epsilon/\sqrt{\alpha}$, while the pruned network will have fewer edges. Moreover, sampling some edges with a probability higher than the $p_{ij}$'s will not harm the bound in Eq. 10 (and will simply increase the number of preserved edges linearly in the degree of oversampling). In particular, any subset of the probabilities can be set to 1; thus the pruning rule can be naturally applied only to a subset of connections. For more details on these arguments see SI S1.4.

**Near-diagonally dominant networks**   Given a matrix $A$ with eigenvalues $\lambda_i$ and some constant $\gamma$, note that the matrix $A_\gamma = A + \gamma I$ has eigenvalues $\lambda_i + \gamma$ and the same eigenvectors as $A$. If $A$ is not diagonally-dominant, the application of noise-prune to $A$ can be analyzed by considering its effect on $A_\gamma$, with $\gamma$ chosen large enough that $A_\gamma$ is diagonally-dominant. There are two additional sources of error in the analysis: first, the probabilities are derived from the covariance matrix of $A$ and are thus sub-optimal for $A_\gamma$; second, the approximation of Eq. 10 holds for $A_\gamma$ with some $\epsilon$ and the corresponding equation for $A$ includes an additive term of magnitude $\epsilon\gamma$ (see SI S2.1 for details).

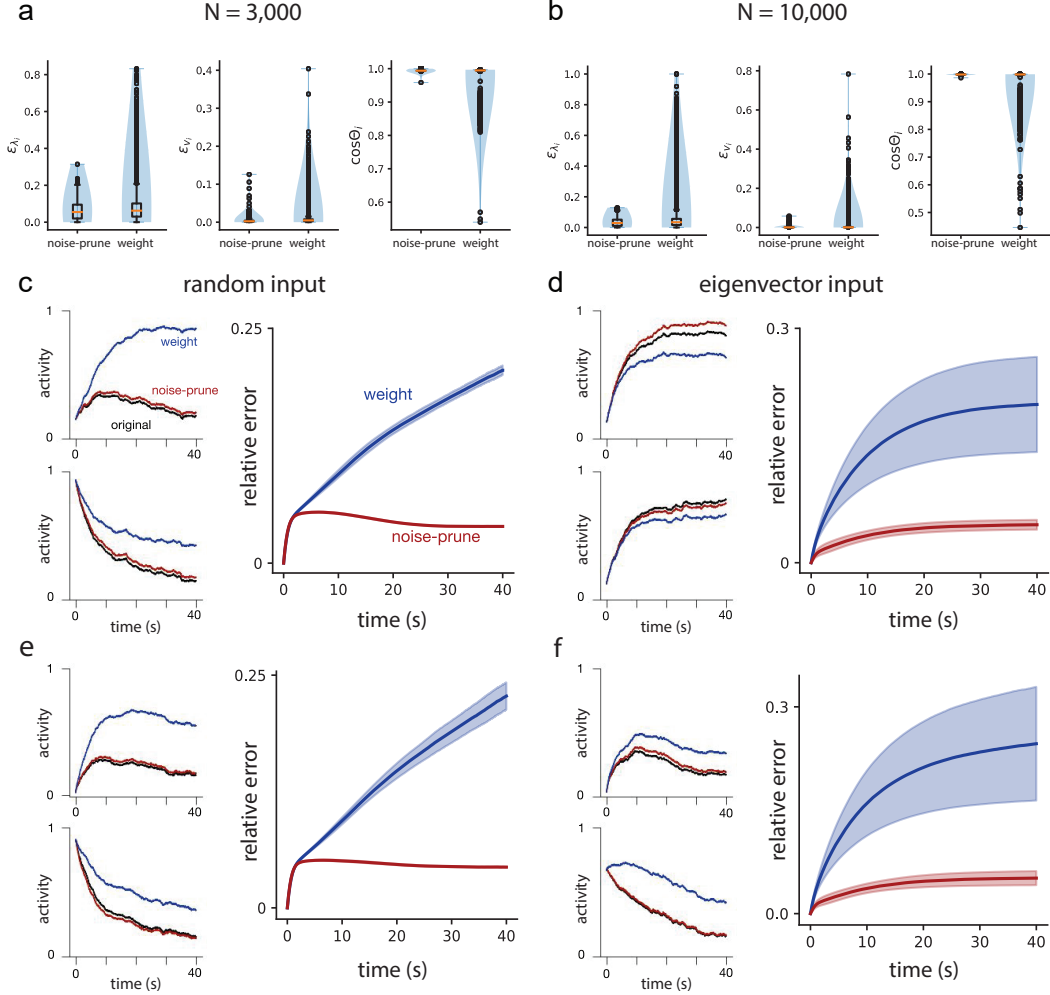

Figure 2: **Noise-prune performance on clustered symmetric and non-symmetric networks.** (a) Performance of noise-prune (left box in each panel) and weight-based pruning (right box in each panel) on networks pruned to $10\%$ density. The left network of size $N = 3,000$ contains 3 clusters of size 100 and 1 cluster of size 2700, with dense within-cluster connections ($60\%, \sim N(1,1)$) and sparse long-range connections ($5000$ total, $\sim U(0,1)$). From left to right, panels show distribution of $\epsilon_{\lambda_i}$, $\epsilon_{v_i}$ and $\cos(\theta_i)$ (defined in text). Note that good performance corresponds to $\epsilon_{\lambda_i}$ and $\epsilon_{v_i}$ near 0 and $\cos(\theta_i)$ near 1. Boxes show upper and lower quartiles, filled circles show outliers, violin plots show density estimate. (b) As in (a) but for larger clustered network ($N = 10,000$, contains 10 clusters of size 100 and 1 cluster of size 9000). (c-f) Dynamical response for networks with three clusters ($1000, 200,$ and $800$ nodes; connections distributed as in (a)). Black traces are the original unpruned network; red traces are networks pruned to $20\%$ sparsity with noise-prune in the matched diagonal setting; blue traces are networks pruned to $20\%$ sparsity using probabilities depending solely on weights. (c) Response of symmetric clustered network to random inputs. Panel shows trajectories from dynamical system $\frac{d\boldsymbol{x}}{dt} = A\boldsymbol{x} + \boldsymbol{b} + \boldsymbol{\xi}(t)$, where $\boldsymbol{b}$ is a small constant background input ($0.0002$), $\boldsymbol{\xi}(t)$ is gaussian white noise, and the initial condition $\boldsymbol{x}(0)$ is chosen with uniformly random entries $U(0,1)$. Left: response of two sample neurons for the three conditions. Right: mean (lines) and standard deviation (shaded area) of relative errors $||\boldsymbol{x}_{orig}(t) - \boldsymbol{x}_{np}(t)||_2/||\boldsymbol{x}_{orig}(t)||_2$ (red) and $||\boldsymbol{x}_{orig}(t) - \boldsymbol{x}_w(t)||_2/||\boldsymbol{x}_{orig}(t)||_2$ (blue) over 20 different initial conditions and pruning runs. Here $\boldsymbol{x}_{orig}, \boldsymbol{x}_{np}, \boldsymbol{x}_w$ are dynamical responses of the original, noise-pruned, and weight-pruned network respectively. (d) As in (c), but with $\boldsymbol{b} = \boldsymbol{x}(0) = \boldsymbol{v}_i$ where $\boldsymbol{v}_i$ is the eigenvector corresponding to the $i$th largest eigenvalue of $A$ (or, equivalently, the $i$th smallest eigenvalue of $B$). Results averaged over 20 slowest eigenvectors ($i = 1, \ldots, 20$). (e), (f) Analogous to (c), (d) respectively but for networks with non-symmetric connections.

**Rectified linear units**   Let $[\cdot]_+ = \max[0, \cdot]$ be a rectified linear activation function and consider the recurrent neural network

$$\frac{d\boldsymbol{x}}{dt} = -D\boldsymbol{x} + [W\boldsymbol{x} + \boldsymbol{b}(t)]_+ . \tag{13}$$

As before, define $A = -D + W$. Let $A^{sparse}$ be the result of applying noise-prune to $A$ using the probabilities from the linear network defined by $A$ (consequently Eq. 3 holds for $A$, $A^{sparse}$).

Let $\Gamma(t) = \{i : \sum_j W_{ij}x_j + b_j(t) > 0\}$ be the indices of neurons that receive suprathreshold input at time $t$. Define $A_{\Gamma(t)}$ and $A_{\Gamma(t)}^{sparse}$ to be the submatrices produced by removing the rows and columns of $A$ and $A^{sparse}$ corresponding to indices not in $\Gamma$. The dynamics of the network in Eq. 13 is approximately determined by the set of linear systems with coupling matrices $A_{\Gamma(t)}$, $A_{\Gamma(t)}^{sparse}$ (proved in SI S2.2). Here, we note that the approximation Eq. 3 for $A$, $A^{sparse}$ implies the same approximation for $A_{\Gamma(t)}$ and $A_{\Gamma(t)}^{sparse}$. Specifically, given some $\Gamma(t)$ with size $|\Gamma(t)|$, let $\Gamma(t, j)$ be the index of the $j$-th active neuron. Now given $\boldsymbol{y} \in \mathbb{R}^{|\Gamma(t)|}$, define a corresponding $\boldsymbol{x} \in \mathbb{R}^N$ as $\boldsymbol{x}(\Gamma(t, j)) = y(j)$ and remaining entries 0. Then $\boldsymbol{y}^T A_{\Gamma(t)}^{sparse} \boldsymbol{y} = \boldsymbol{x}^T A^{sparse} \boldsymbol{x}$, and similarly for $A_{\Gamma(t)}$ and $A$. Substituting into Eq. 3 shows that the approximation holds for $A_{\Gamma(t)}$, $A_{\Gamma(t)}^{sparse}$.

The argument requires sampling probabilities computed from the covariance matrix of the dynamical system with coupling matrix $A$. A simple way to determine these is to add non-specific background excitation or global fluctuations in excitability to the network to push neurons away from the threshold. Intriguingly, such global excitability fluctuations are observed during slow-wave sleep [26, 27].

# 7   Discussion

The structure of the sampling argument, the notion of spectral approximation, and the use of matrix concentration of measure tools are drawn from a rich body of work on graph sparsification [16–18], particularly the beautiful paper of Spielman & Srivastava (2011). Our study links these results with neural networks and noisy dynamical systems. In the graph Laplacian context, the counterpart of the diff-cov matrix (see Eq. 4) is "effective resistance", which measures the electrical resistance between nodes if the graph is considered a weighted resistor network. Effective resistance has multiple nice properties [28, 16], such as forming a natural metric [29], and the diff-cov matrix may be similarly useful for neural networks. Conversely, a difference of covariances has recently been suggested to generalize effective resistance to directed graphs [30]. There may be further useful connections to be drawn between this set of ideas and noise-driven dynamics in neural networks.

A number of studies have investigated task-dependent pruning of connections in artificial neural networks, often with very compelling results [31–41]. Current state-of-the-art approaches in machine learning typically train a network to good performance on a task, assign a measure of importance to each connection in the network (often its weight and sometimes a measure of impact on the task cost function such as terms in the Hessian), remove connections from the network according to this importance measure, and then repeat the cycle of training and pruning (see [41] for a recent review). Such approaches have been extremely successful, yielding networks with greatly reduced density (as little as a few percent of the original) while preserving task performance. Our work is complementary to these pruning studies in three ways. First, these studies focus on the supervised, (typically) feedforward setting, and algorithms are not usually biologically plausible. By contrast, the current study seeks an unsupervised, biologically-plausible algorithm for recurrent networks. Second, most existing studies typically seek good empirical performance in quite challenging real-world applications rather than theoretical results, while we focus on developing strong theoretical results in a limited setting. And finally, existing algorithms that prune connections typically do so either based on connection weight or a nonlocal measure of cost function sensitivity, while we combine weight with a local term that extracts a connection's importance to the network from activity fluctuations. Our study is most reminiscent of unsupervised approaches that merge or remove highly correlated neurons [42–44], though the setting, algorithms and theoretical guarantees are quite different, and we consider weight pruning rather than removing entire neurons. Note that we do not expect noise-prune to be competitive with state-of-the-art supervised approaches in machine learning when measured by preserving performance on a given task (rather than preserving dynamics). However, the novel perspective provided by noise-prune and the theoretical results may be useful in developing more powerful algorithms for task-driven pruning.

The proofs apply to the limited case of symmetric diagonally-dominant linear and rectified linear recurrent networks. Certain networks in the brain may potentially be modeled as diagonally-dominant (e.g., in the high-conductance regime, when membrane time constants are very small [45]), though it is unclear how good this approximation will be. More importantly, connections in biological neural networks are not symmetric. The framework may apply more naturally to excitatory (or inhibitory) sub-networks with a higher probability of reciprocal connectivity [46], and especially to cell assemblies that code for the same stimulus or concept [47]. Finally, biological networks are nonlinear. Thus, the theoretical framework presented is far from general.

However, we highlight two causes for optimism. First, in the limited regime where the theory applies, results are very strong and robust (as in graph Laplacian sparsification [16, 17]), able to asymptotically preserve the entire spectrum even when the fraction of retained edges goes to 0. Preservation of the entire spectrum is likely too strong for neural networks, which often show redundant coding and low-dimensional dynamics. It may be possible to more weakly approximate a broader family of networks. Second, the noise-prune rule itself (Eq. 4) does not require particular network structure and can in principle be applied to any recurrent network (note that covariance for a general normal matrix is determined by the symmetric part of the matrix). Indeed, we empirically find that noise-prune preserves dynamics in non-symmetric clustered networks, Fig. 2e, f, and thus shows good performance beyond the regime where theoretical guarantees hold. A more exhaustive empirical characterization of noise-prune is beyond the scope of the present study, but this is a natural direction for future work.

The pruning rule uses randomness twice. First, it uses noisy fluctuations in activity to probe network structure and make global information locally available in the form of activity correlations between pairs of neurons. Second, it randomly decides whether to preserve and strengthen or prune a connection. This use of randomness is inspired by seemingly ubiquitous noise at multiple levels in neural systems [12–15]. It is still unclear how much of this "noise" reflects the encoding of unknown variables as opposed to genuine randomness, and to what degree noise is averaged away as opposed to being used as a computational resource. However, randomized algorithms are often appealingly simple, powerful and easy to parallelize, and it is plausible and widely speculated that brains have evolved to take computational advantage of biological noise [14, 48].

Unlike pruning rules that remove (typically weak) synapses and simply preserve the others, the (subset of) synapses targeted by noise-prune are either removed or strengthened, reminiscent of observations that small spines on neurons are highly variable and liable to either vanish or grow and stabilize [49]. More generally, a strengthen-or-prune rule like that in Eq. 5 can be applied with different sampling probabilities, which may be appropriate for different settings, and synapses can be strengthened or weakened rather than pruned. If weights and probabilities are chosen to preserve synaptic weights on average (which is a natural target for an unsupervised algorithm), then the approach approximately preserves total synaptic input to and output from a neuron as well as the dynamics resulting from a given input or network activity state. The theoretical approach may thus be more generally useful in settings where synaptic weight is redistributed across synapses (such as in some homeostatic mechanisms [23]).

In this study we have focused on pruning synapses while preserving existing network dynamics, thus approaching pruning primarily as resource conservation. Pruning in the brain may serve other functions as well, such as making networks faster or more robust to noise. Given that the pruned network needs to carry out a similar set of input-output transformations to the original network, dynamical patterns are likely to be similar between unpruned and pruned networks and thus preservation of dynamics such as proposed here could be used as a building block to investigate more complex pruning algorithms that optimize other features of network responses.

The approach presented here suggests decomposing into two pieces the difficult problem of learning a sparse network solution to a task. First, a greedy task-driven learning epoch that adds synapses where they might be needed, regardless of efficiency (such as would be expected from correlational / Hebbian learning processes). And second, a noisy, task-agnostic, anti-Hebbian epoch during which a subset of synapses enter a labile state and are either consolidated or pruned. The second regime is reminiscent of theories of sleep [50, 51] and it will be interesting to attempt to connect sleep phenomenology with the algorithm presented in this study.

## Broader Impact

While the larger question that motivates this study (how the brain might prune synapses) is of great practical interest, the results presented here are purely theoretical and quite abstract, and we do not foresee any immediate societal consequences or ethical issues.

## Acknowledgments and Disclosure of Funding

We thank E. Natale, C. Papadimitriou and L. Turin for helpful discussions on graph sparsification and A. Bernacchia, K. Mimmack and E. Natale for comments on the manuscript. Part of this work was conceived when RC was a Google Research Fellow at the Simons Institute for the Theory of Computing at UC Berkeley. EM was partially supported by a UC Davis Summer GSR Award for Engineering or Computer-related Applications and Methods. The authors were benefited by participating in the activities of the UC Davis TETRAPODS Institute of Data Science, which has been funded by the NSF TRIPODS grant CCF-1934568.

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
