[Supplementary Material]

# Supplementary Information

In this section we expand on the arguments in the main text. Note that, for completeness, some portions of the main text are repeated here.

## Contents

## S1    Theoretical framework underlying noise-prune

We consider the $N \times N$ coupling matrix of the linear system

$$\frac{dx}{dt} = Ax + \boldsymbol{b}(t), \tag{1}$$

and describe how to construct a sparse matrix $A^{sparse}$ whose spectrum (and hence dynamics) are similar to $A$.

To measure the similarity of $A$ and $A^{sparse}$, we adopt the notion of spectral similarity [1, 2] from the field of graph sparsification and require that for some small $\epsilon > 0$,

$$|\boldsymbol{x}^T(A^{sparse} - A)\boldsymbol{x}| \leq \epsilon|\boldsymbol{x}^T A\boldsymbol{x}| \quad \forall \boldsymbol{x} \in \mathbb{R}^N. \tag{2}$$

The primary theoretical insights of this section are that (a) results on the sparsification of graph Laplacians [1, 2] can be applied, with slight generalization, to pruning signed symmetric diagonally-dominant linear neural networks and (b) that the covariance matrix of the network when driven by noise provides appropriate pruning probabilities. We also discuss what properties of the original network are preserved after sparsifying the matrix $A$, as well as how these maintained properties are affected when the sampling probabilities are changed.

## S1.1    Sparsification of symmetric, diagonally-dominant networks

In this section we show how to construct spectral sparsifiers of $A$. We follow the proof of [1], with some adaptation.

Let $A$ be the coupling matrix of a linear system, as in Eq. 1. Note that in order for the linear system to be stable, all the eigenvalues of $A$ must have negative real part (and

hence the matrix must be invertible). A non-invertible coupling matrix would correspond to a network with an unrealistically long (i.e., infinite) time-constant.

We impose the further restrictions that $A$ be a symmetric, *diagonally-dominant* matrix; that is, $A_{ij} = A_{ji}$ and $|A_{ii}| \geq \sum_{j \neq i} |A_{ij}|$. In the main text, we focused on the case where this inequality was saturated (i.e., $|A_{ii}| = \sum_{j \neq i} |A_{ij}|$). Here, we expand the proof to include a *strictly diagonally-dominant* $A$, thus satisfying $|A_{ij}| > \sum_{j \neq i} |A_{ij}|$ (note that the argument is essentially the same and also that both the original and matched diagonal cases of noise-prune simply preserve any excess weight along the diagonal). The diagonal entries of $A$ reflect the intrinsic leak of activity and are negative. Combined with the strict diagonal-dominance requirement, the negative diagonal also implies that the eigenvalues of $A$ are negative, as can be seen from, e.g., a Gershgorin disk argument. Note that the diagonal dominance condition is stronger than the requirement of negative eigenvalues. In the event that $A$ satisfies $|A_{ii}| = \sum_{j \neq i} |A_{ij}|$ as in the main text, invertibility is no longer guaranteed by the Gershgorin disk argument but we assume invertibility based on the stability of the equivalent linear system. We can also relax the invertibility condition by considering the pseudoinverse of $A$ and working in the subspace orthogonal to the nullspace of $A$ (as done for graph Laplacians [1]). To sum up, $A$ is negative definite since it is symmetric with negative eigenvalues.

For notational convenience, set $B = -A$ (and note that $B$ is positive definite). The non-zero off-diagonal entries $b_{ij} = -a_{ij} = -w_{ij}$ correspond to the connections in the network (note that throughout $w_{ij}$ refers to the weight in $A$, i.e., $-b_{ij}$; the argument can be rewritten without introducing $B$ at the cost of extra minus signs).

**Edge decomposition** For each of these undirected connections $(i,j)$ with $i > j$, we define the edge matrix $X^{(i,j)}$ by

$$X_{k\ell}^{(i,j)} = \begin{cases} |w_{ij}| & \text{if } (k,l) = (i,i) \text{ or } (j,j) \\ -w_{ij} & \text{if } (k,l) = (i,j) \text{ or } (j,i) \\ 0 & \text{otherwise.} \end{cases} \tag{3}$$

Note that there is no restriction on the sign of $w_{ij}$. Also notice that $X^{(i,j)}$ can be written as $\boldsymbol{v}_{ij} \boldsymbol{v}_{ij}^T$ where $\boldsymbol{v}_{ij} \in \mathbb{R}^N$ has $i$th entry $\sqrt{|w_{ij}|}$ and $j$th entry $-\text{sgn}(w_{ij})\sqrt{|w_{ij}|}$. Thus $X^{(i,j)}$ a rank-1 matrix. Moreover, since the non-zero eigenvalue is positive, the matrix is positive semidefinite. Also note that specifying $i > j$ above is simply a manner of convention to not double-count connections in the symmetric matrix.

We also define the matrix $X^{(i,i)}$ for all $i$ to have only a single non-zero entry $X_{ii}^{(i,i)} = B_{ii} - \sum_{j \neq i} |w_{ij}|$. Because $B$ is diagonally-dominant with positive diagonal, the single non-zero entry of $X^{(i,i)}$ is positive, again implying that $X^{(i,i)}$ is rank-1 positive semidefinite. We include these diagonal pieces in the sampling argument for completeness but will usually simply treat them as fixed.

The original matrix $B$ is the sum of these edge matrices, $B = \sum_{i \geq j} X^{(i,j)}$ (where the notation $\sum_{i \geq j}$ sums over all existing edge pairs where $i \geq j$).

**Sampling edges** Now, for $i \geq j$, define the random matrix $\tilde{X}^{ij}$ as

$$\tilde{X}^{ij} = \begin{cases} X^{(i,j)}/p_{ij} & \text{with probability } p_{ij} \\ 0 & \text{otherwise,} \end{cases} \tag{4}$$

where $0 < p_{ij} \leq 1$ is some probability we will determine below. Observe that, regardless of the choice of the $p_{ij}$'s, $\mathbb{E}[\tilde{X}^{ij}] = p_{ij}X^{(i,j)}/p_{ij} = X^{(i,j)}$. Correspondingly, for any set of probabilities, $\boldsymbol{p}$, we can define the matrix $B^{sparse,\boldsymbol{p}} = \sum_{i \geq j} \tilde{X}^{ij}$ and we have $\mathbb{E}[B^{sparse,\boldsymbol{p}}] = B$ (note that $B^{sparse,\boldsymbol{p}}$ will only be sparse if the $p_{ij}$ are small).

**Transformation to identity** Analogous to Spielman & Srivastava (2011), we implement their argument in our framework by first transforming $B$ into the identity matrix $I$ and finding an appropriate approximation $\tilde{I}$, with the goal of transforming back and arriving at our desired sparsifier $B^{sparse}$. This step is crucial for preserving the entire spectrum (as required by Eq. 2), rather than only the largest eigenvalue (and leads to the diff-cov term in the probabilities).

First observe that $I = B^{-1/2}BB^{-1/2}$, where $B^{-1/2}$ is the matrix whose square is $B^{-1}$ ($B^{-1/2}$ exists since $B$ is invertible and diagonalizable and moreover is real-valued since $B$ is positive definite).[a] Then, defining $Y^{(i,j)} = B^{-1/2}X^{(i,j)}B^{-1/2}$, we have

$$I = B^{-1/2}BB^{-1/2} = \sum_{i \geq j} B^{-1/2}X^{(i,j)}B^{-1/2} = \sum_{i \geq j} Y^{(i,j)}. \tag{5}$$

This gives motivation to define the random matrices $\tilde{Y}^{ij} = B^{-1/2}\tilde{X}^{ij}B^{-1/2}$ and $\tilde{I} = \sum_{i \geq j} \tilde{Y}^{ij}$. Note that $E(\tilde{I}) = I$.

Now, for given $0 < \epsilon < 1$, our goal will be to choose $p_{ij}$ in order to guarantee that

$$\boldsymbol{y}^T(1-\epsilon)I\boldsymbol{y} \leq \boldsymbol{y}^T\tilde{I}\boldsymbol{y} \leq \boldsymbol{y}^T(1+\epsilon)I\boldsymbol{y} \quad \forall \boldsymbol{y} \in \mathbb{R}^N, \tag{6}$$

with high probability (w.h.p.). If we can do so, then for a given $\boldsymbol{x} \in \mathbb{R}^N$, we can set $\boldsymbol{y} = B^{1/2}\boldsymbol{x}$ in order to arrive at, w.h.p.,

$$\boldsymbol{x}^T(1-\epsilon)B\boldsymbol{x} \leq \boldsymbol{x}^T B^{sparse}\boldsymbol{x} \leq \boldsymbol{x}^T(1+\epsilon)B\boldsymbol{x} \quad \forall \boldsymbol{x} \in \mathbb{R}^N, \tag{7}$$

where $B^{sparse} = B^{-1/2}\tilde{I}B^{-1/2} = \sum_{i \geq j} \tilde{X}^{ij}$, which provides the desired approximation.

**Probabilities from matrix Chernoff bound** We want our $p_{ij}$ to be as small as possible while still maintaining the inequalities Eq. 6, 7. To derive good choices for the $p_{ij}$'s, we apply the matrix Chernoff bound [3–6] to bound the fluctuations of $\tilde{I} = \sum_{i \geq j} \tilde{Y}^{ij}$ around its expectation value, $I$. Let $M$ be an upper bound on the $\tilde{Y}^{ij}$'s, so that $0 \leq ||\tilde{Y}^{ij}||_2 \leq M$. Let $\lambda_{min}$ and $\lambda_{max}$ indicate minimum and maximum eigenvalues. The bound then guarantees that

$$P\left[\lambda_{min}\left(\sum_{i \geq j} \tilde{Y}^{ij}\right) \leq (1-\epsilon)\right] \leq N\left(\frac{e^{-\epsilon}}{(1-\epsilon)^{(1-\epsilon)}}\right)^{1/M} \leq Ne^{-\epsilon^2/2M} \quad \text{for } 0 < \epsilon < 1,$$

$$P\left[\lambda_{max}\left(\sum_{i \geq j} \tilde{Y}^{ij}\right) \geq (1+\epsilon)\right] \leq N\left(\frac{e^{\epsilon}}{(1+\epsilon)^{(1+\epsilon)}}\right)^{1/M} \leq Ne^{-\epsilon^2/3M} \quad \text{for } 0 < \epsilon \tag{8}$$

---

[a]If the eigenvector decomposition of $B$ is $UDU^{-1}$ then $B^{-1/2}$ can be constructed as $UD^{-1/2}U^{-1}$, where the entries of $D^{-1/2}$ are the inverse square roots of the corresponding entries of $D$).

The hypothesis of the bound requires the spectral norm of the $\tilde{Y}^{ij}$'s to be uniformly bounded across all edges; i.e., $||\tilde{Y}^{ij}|| \leq M$. Moreover $||\tilde{Y}^{ij}||$ depends on $1/p_{ij}$, so smaller probabilities lead to a larger bound $M$. Thus we choose the $p_{ij}$ in order to minimize $M$.

Since $||\tilde{Y}^{ij}||$ is either $\frac{1}{p_{ij}}\left\|Y^{(i,j)}\right\|$ or 0, choose $p_{ij}$ to equalize the upper bound on $||\tilde{Y}^{ij}||$ across all $i \geq j$:

$$p_{ij} = K_{deg}\left\|Y^{(i,j)}\right\| = K_{deg}||B^{-1/2}X^{(i,j)}B^{-1/2}|| \tag{9}$$

where $K_{deg}$ is some constant. This guarantees that $||\tilde{Y}^{ij}|| \leq M = 1/K_{deg}$. Thus, if we take $K_{deg} \geq 4\log(N)/\epsilon^2$, the probabilities in Eq. 8 are guaranteed to be smaller than $1/N$ and $1/N^{1/3}$, respectively. Consequently, this choice of probabilities guarantees that Eqs. 6, 7 are satisfied w.h.p., as desired.

Note that the constant 4 is chosen somewhat arbitrarily here, with a larger constant corresponding to faster-decaying probabilities in Eq. 8 but also a larger number of edges expected to be sampled (since each $\tilde{Y}^{ij}$ is less likely to take on the value of 0).

**Bound on number of edges** Since edge $(i, j)$ is independently included with probability $p_{ij}$, the expected number of edges in the network is $\langle N_{edges}\rangle = \sum_{i>j} p_{ij}$ (note the strict inequality here, as $i = j$ does not correspond to edges, but rather the leak in neuronal activity).

We have

$$\sum_{i>j} p_{ij} \leq \sum_{i \geq j} p_{ij} = K_{deg}\sum_{i \geq j}\left\|Y^{(i,j)}\right\| = K_{deg}\sum_{i \geq j}\left\|B^{-1/2}X^{(i,j)}B^{-1/2}\right\| \tag{10}$$

Note that $Y^{(i,j)} = \boldsymbol{u}_{ij}\boldsymbol{u}_{ij}^T$, where $\boldsymbol{u}_{ij} = B^{-1/2}\boldsymbol{v}_{ij}$ and $\boldsymbol{v}_{ij}$ is the vector defined after Eq. 3. Consequently, $Y^{(i,j)}$ is rank-1 with positive eigenvalue and $\left\|Y^{(i,j)}\right\| = \operatorname{tr} Y^{(i,j)}$. This yields

$$\sum_{i \geq j}\left\|Y^{(i,j)}\right\| = \sum_{i \geq j}\operatorname{tr} Y^{(i,j)} = \operatorname{tr}\left(B^{-1/2}\sum_{i \geq j}X^{(i,j)}B^{-1/2}\right) = \operatorname{tr}(B^{-1/2}BB^{-1/2}) = \operatorname{tr}(I) = N. \tag{11}$$

Thus we have

$$\langle N_{edges}\rangle = \sum_{i>j} p_{ij} = \sum_{i>j} K_{deg}\left\|Y^{(i,j)}\right\| \leq NK_{deg}. \tag{12}$$

**Simple expression for probabilities** Note that $||B^{-1/2}X^{(i,j)}B^{-1/2}|| = \operatorname{tr}\left(B^{-1/2}X^{(i,j)}B^{-1/2}\right) = \operatorname{tr}\left(B^{-1}X^{(i,j)}\right)$, again using the fact that the trace of a positive semi-definite rank-1 matrix is its spectral norm, and that the trace is cyclic (and $B^{-1/2}B^{-1/2} = B^{-1}$ by definition). The product $B^{-1}X^{(i,j)}$ has only two non-zero diagonal terms: its $i$th diagonal element is given by $|w_{ij}|B_{ii}^{-1} - w_{ij}B_{ij}^{-1}$ and its $j$th diagonal element is given by $-w_{ij}B_{ji}^{-1} + |w_{ij}|B_{jj}^{-1}$. Using the trivial decomposition $w_{ij} = \operatorname{sgn}(w_{ij})|w_{ij}|$ and adding these two diagonal elements together, we see that

$$p_{ij} = K_{deg}\operatorname{tr}\left(B^{-1}X^{(i,j)}\right) = K_{deg}|w_{ij}|(B_{ii}^{-1} + B_{jj}^{-1} - \operatorname{sgn}(w_{ij})2B_{ij}^{-1}), \tag{13}$$

where we note that $B_{ij}^{-1} = B_{ji}^{-1}$, since the inverse of a symmetric matrix is symmetric.

Similarly, the $p_{ii}$ are observed to be

$$p_{ii} = K_{deg}||B^{-1/2}X^{(i,i)}B^{-1/2}|| = K_{deg}\operatorname{tr}\left(B^{-1/2}X^{(i,i)}B^{-1/2}\right) = K_{deg}\operatorname{tr}\left(B^{-1}X^{(i,i)}\right) \tag{14}$$

where we again use the cyclic property of the trace. Since the product $B^{-1}X^{(i,i)}$ has only the single non-zero diagonal element $B_{ii}^{-1}(B_{ii} - \sum_{j \neq i} |w_{ij}|)$, we arrive at the simple expression $p_{ii} = K_{deg}B_{ii}^{-1}\left(B_{ii} - \sum_{j \neq i} |w_{ij}|\right)$. Note that in practice we simply set this probability to 1, but include it here for completeness.

Finally, recall that $B = -A$ and note that $A^{sparse} = -B^{sparse}$ is the outcome of the pruning applied to $A$. Substituting for $B$ in terms of $A$, the sampling probabilities are

$$p_{ij} = -K_{deg}|w_{ij}|(A_{ii}^{-1} + A_{jj}^{-1} - \text{sign}(w_{ij})2A_{ij}^{-1}) \tag{15}$$

## S1.2 Sampling probabilities from noise-driven covariance

The matrix inverse term $-A^{-1}$ in Eq. 15 has a natural interpretation in terms of the covariance matrix of the corresponding linear dynamical system when driven by white noise. When the network is driven by noise, the dynamics are

$$\frac{dx}{dt} = A\boldsymbol{x} + \sigma\boldsymbol{\xi}(t), \tag{16}$$

where $\boldsymbol{\xi}$ is unit variance Gaussian white-noise at each neuron and $\sigma$ is the standard deviation of the noise (note that this is a stochastic differential equation).

The covariance matrix of the resulting dynamics is given as the solution to the Lyapunov equation [7, 8]:

$$AC + CA^* = -\sigma^2 I. \tag{17}$$

Assume that $A$ is normal, meaning that $A^*A = AA^*$, where $A^*$ is the conjugate transpose of $A$. Note that all symmetric matrices are normal. Since $A$ is normal it can be diagonalized as $A = U\Lambda U^*$, where $\Lambda$ is a diagonal matrix of eigenvalues and $U$ is unitary.

Substituting the decomposition of $A$ into Eq. 17 we have

$$-\sigma^2 I = U\Lambda U^*C + CU\Lambda^*U^* \tag{18}$$

so that multiplying this equation through by $U^*$ on the left and $U$ on the right and defining $\tilde{C} = U^*CU$, we arrive at

$$-\sigma^2 I = \Lambda U^*CU + U^*CU\Lambda^* = \Lambda\tilde{C} + \tilde{C}\Lambda^*. \tag{19}$$

Since $\Lambda$ is diagonal, the equation can be solved for the entries of $\tilde{C}$. $\tilde{C}$ is diagonal, with diagonal entries $\tilde{C}_{ii} = -\frac{\sigma^2}{\lambda_i + \lambda_i^*}$, where $\lambda_i$ and $\lambda_i^*$ are the $i$-th diagonal entries of $\Lambda$ and $\Lambda^*$ respectively (i.e., the $i$-th eigenvalue of $A$). By definition $C = U\tilde{C}U^*$ and thus $C$ has the same eigenvectors as $A$, with eigenvalues given by the diagonal entries of $\tilde{C}$.

Define the symmetric part of $A$ to be $A_{symm} = \frac{1}{2}(A + A^*)$ and observe that this has eigenvalues $\frac{1}{2}(\lambda_i + \lambda_i^*)$. Thus, $C = -\frac{\sigma^2}{2}A_{symm}^{-1}$. In particular, for the symmetric matrices considered in the previous section, $C = -\frac{\sigma^2}{2}A^{-1}$. Substituting into the theoretically-derived form for the sampling rule and absorbing $\frac{\sigma^2}{2}$ into the overall constant yields

$$p_{ij} = K|w_{ij}|(C_{ii} + C_{jj} - \text{sign}(w_{ij})2C_{ij}) \tag{20}$$

## S1.3 What is preserved

The notion of spectral sparsification that we adopt from the graph Laplacian literature [2, 1] (see Eq. 2) is quite strong and here we briefly discuss some of the properties it entails.

Recall that, given $0 < \epsilon < 1$, Eq. 7 guarantees that

$$\boldsymbol{x}^T(1 - \epsilon)B\boldsymbol{x} \leq \boldsymbol{x}^T B^{sparse}\boldsymbol{x} \leq \boldsymbol{x}^T(1 + \epsilon)B\boldsymbol{x} \quad \forall \boldsymbol{x} \in \mathbb{R}^N, \tag{21}$$

so that substituting $A = -B$ and rearranging yields the approximation from the main text

$$|\boldsymbol{x}^T(A^{sparse} - A)\boldsymbol{x}| \leq \epsilon|\boldsymbol{x}^T A\boldsymbol{x}| \quad \forall \boldsymbol{x} \in \mathbb{R}^N, \tag{22}$$

where we use the fact that $A$ is negative definite to see that $-\boldsymbol{x}^T A\boldsymbol{x} = |\boldsymbol{x}^T A\boldsymbol{x}|$.

By definition, Eq. 22 approximately preserves $A$ as a quadratic form and thus apart from the eigenvalues and products described below, it also preserves properties of the dynamical system that depend on $A$ as a quadratic form, such as the resting state variances, the diagonal elements of A and the differences-of-covariances (diff-covs).

**Eigenvalues** Let $\lambda_1 \leq \lambda_2 \leq \cdots \leq \lambda_N$ and $\tilde{\lambda}_1 \leq \tilde{\lambda}_2 \leq \cdots \leq \tilde{\lambda}_N$ be the eigenvalues of $B$ and $B^{sparse}$ respectively.

Let $S$ denote the collection of subspaces $U \subset \mathbb{R}^N$ with $\dim U = k$, and consider the functions $f_B, f_{B^{sparse}} : S \to \mathbb{R}$ given by

$$f_B(U) = \max_{\substack{\boldsymbol{x} \in U \\ \|\boldsymbol{x}\|=1}} \boldsymbol{x}^T B\boldsymbol{x}, \qquad f_{B^{sparse}}(U) = \max_{\substack{\boldsymbol{x} \in U \\ \|\boldsymbol{x}\|=1}} \boldsymbol{x}^T B^{sparse}\boldsymbol{x}. \tag{23}$$

Let $U \in S$ be a given subspace of $\mathbb{R}^N$ with dimension $k$. Since $(1-\epsilon)\boldsymbol{x}^T B\boldsymbol{x} \leq \boldsymbol{x}^T B^{sparse}\boldsymbol{x} \leq (1+\epsilon)\boldsymbol{x}^T B\boldsymbol{x}$ for all $\boldsymbol{x} \in \mathbb{R}^N$, we can take the maximum over all $\boldsymbol{x} \in U \subset \mathbb{R}^N$ with unit norm to see that

$$(1 - \epsilon)f_B(U) \leq f_{B^{sparse}}(U) \leq (1 + \epsilon)f_B(U). \tag{24}$$

Since this inequality holds for any subspace, taking a minimum over all subspaces in $S$ still preserves the inequality:

$$(1 - \epsilon)\min_{U \in S} f_B(U) \leq \min_{U \in S} f_{B^{sparse}}(U) \leq (1 + \epsilon)\min_{U \in S} f_B(U). \tag{25}$$

Thus, by the Courant-Fischer Theorem, we arrive at

$$(1 - \epsilon)\lambda_k \leq \tilde{\lambda}_k \leq (1 + \epsilon)\lambda_k, \quad \forall 1 \leq k \leq N \tag{26}$$

Thus all eigenvalues are preserved within a multiplicative factor of $\epsilon$.

**Eigenvectors** Here we show that the angle between eigenvectors is preserved up to a factor depending on the arbitrarily small degree of spectral approximation $\epsilon$. First, note that a rearrangement of the Davis-Kahan theorem states

$$\sqrt{1 - \frac{4\|A - A^{sparse}\|^2}{\delta_i^2}} \leq \cos \angle(v_i, \tilde{v}_i), \qquad \text{for all } i \tag{27}$$

where $v_i$ and $\tilde{v}_i$ are the $i$th eigenvectors of $A$ and $A^{sparse}$ respectively, and

$$\delta_i = \min_{j:j \neq i} |\lambda_i(A) - \lambda_j(A)| > 0. \tag{28}$$

Fix $\gamma > 0$. Now, setting $\epsilon = \frac{\gamma}{2\lambda_{max}}$ and constructing the corresponding $A^{sparse}$, we know

$$||A - A^{sparse}|| = \sup_{||x||=1} |x^T (A - A^{sparse})x| \leq \sup_{||x||=1} \epsilon |x^T A x| = \epsilon \lambda_{max} = \frac{\gamma}{2} \qquad (29)$$

where the first equality holds since $A - A^{sparse}$ is Hermitian. Thus, we have

$$\sqrt{1 - \frac{\gamma^2}{\delta_i^2}} \leq \sqrt{1 - \frac{4||A - A^{sparse}||^2}{\delta_i^2}} \leq \cos \angle(v_i, \tilde{v}_i). \qquad (30)$$

Since $\gamma > 0$ was arbitrary, this quantity can be made arbitrarily close to 1. That is, we can guarantee that corresponding eigenvectors of $A$ and $A^{sp}$ point in nearly the same direction.

**Preserved matrix-vector products** First, note that there exist $N$ linearly independent vectors $\{\boldsymbol{w}_1, \ldots, \boldsymbol{w}_N\}$ (i.e., a basis for $\mathbb{R}^N$) for which the matrix vector products are preserved between $B$ and $B^{sparse}$ (or $A$ and $A^{sparse}$) to within $\epsilon$. Let $\boldsymbol{v}_k$ be an eigenvector of $\tilde{I}$ with eigenvalue $1 + \delta_k$ (note that, from Eq. 26, $|\delta_k| \leq \epsilon$). Define $\boldsymbol{w}_k = B^{-1/2}\boldsymbol{v}_k$ and note that $B\boldsymbol{w}_k = B^{1/2}\boldsymbol{v}_k$. Now $B^{sparse}\boldsymbol{w}_k = B^{1/2}\tilde{I}B^{1/2}\boldsymbol{w}_k = B^{1/2}\tilde{I}\boldsymbol{v}_k = (1+\delta_k)B^{1/2}\boldsymbol{v}_k = (1+\delta_k)B\boldsymbol{w}_k$. Consequently, $||(B - B^{sparse})\boldsymbol{w}_k|| \leq \epsilon ||B\boldsymbol{w}_k||$.

Second, note that the eigenspaces of $B$ and $B^{sparse}$ are close, as we show empirically in Fig. 2 of the main text, though precise bounds will depend on how close the corresponding eigenvalue is to another eigenvalue in the spectrum.

Finally, scalar concentration of measure arguments suggests that a rule of the form in Eq. 4 should preserve dense matrix-vector products, provided the entries in the matrix do not grow too large (as for the matrix concentration of measure case). Note, however, that the products of $B$ and $B^{sparse}$ with sparse vectors may be quite different (as will be true for any sparse matrix approximation), because these products are determined by the sum of only a few entries in $B$ and $B^{sparse}$.

## S1.4 Partial sampling and robustness to changing probabilities

**Oversampling** The Chernoff bound in Eq. 8 depends on the sampling probabilities only through the upper bound on the norm of the edge matrices, requiring $0 \leq ||Y_{ij}|| \leq 1/K_{deg}$. In particular, if the derived $p_{ij}$ for an edge is $< 1$ then the same bound holds for any $\tilde{p}_{ij} \geq p_{ij}$ (see below for $p_{ij} > 1$). Consequently, the probabilities described in Eq. 9 above are a lower bound, for a given desired degree of approximation ($\epsilon$). If some of the edges are sampled with a greater probability than the theoretical result derived, the approximation equation Eq. 9 will still hold (though some of the terms in the sum of Eq. 5 will have norm less than $1/K_{deg}$).

The only consequence of over-sampling synapses is that the number of connections in the pruned network will be greater, but the increase is as well behaved as could be desired, corresponding exactly to the degree of over-sampling as $\langle N_{edges} \rangle = \sum_{i>j} p_{ij}$. Furthermore, there is no harm in setting all of the $p_{ii}$ to 1, as these probabilities do not correspond to edges, but rather the diagonal terms, which relate to the intrinsic leak in the activity of neurons in Eq. 1.

Moreover, as long as the sampling probabilities are above the theoretically-derived bound, they can be chosen completely independently at each synapse and do not need to compensate for each other in any way.

**Misspecified probabilities**  Sampling-based sparsification is quite robust to misspecified sampling probabilities [1]. Again, this robustness emerges because probabilities only affect the Chernoff bound through their effect on the norm of the edge matrices. If some synapses are under-sampled using probability $\hat{p}_{ij} = \alpha p_{ij}$, with $\alpha < 1$, the bound on the $||\tilde{Y}^{ij}||$'s inflates by a factor of $1/\alpha$ and the degree of approximation becomes $\hat{\epsilon} = \epsilon/\sqrt{\alpha}$ while preserving the same bound on the probabilities in Eq. 8 (thus maintaining the likelihood of our approximation occuring w.h.p.). To see this, observe that

$$P\left[\lambda_{min} \leq (1 - \hat{\epsilon})\right] \leq N \left(e^{-\hat{\epsilon}^2/2}\right)^{\alpha/M} = N \left(e^{-\epsilon^2/2}\right)^{1/M}, \tag{31}$$

and similarly for the other inequality in Eq. 8. Note here that we could instead choose to maintain our original degree of approximation $\epsilon$, but this would correspond to the larger upper bound

$$P\left[\lambda_{min} \leq (1 - \epsilon)\right] \leq N \left(e^{-\epsilon^2/2}\right)^{\alpha/M} \tag{32}$$

which means that Eq. 7 would occur with lower probability.

**Fixed edges**  The sampling argument can be applied to only a subset of edges in several ways. A particularly natural approach is to simply set the sampling probability for a fixed edge $\tilde{p}_{ij} = 1$ and note that, if $p_{ij} < 1$ the bound $||\tilde{Y}^{ij}|| \leq M$ still holds and so do the subsequent theoretical results.

A second way to apply the argument to a subset of edges is to write the matrix $A$ as $A_{fixed} + A_{sample}$, where $A_{fixed}$ is the submatrix of edges that are to be preserved and $A_{sample}$ is the submatrix of edges to be either pruned or strengthened. The argument in Section S1.1 can then be applied to $A_{sample}$ (note that $A_{sample}$ is diagonally dominant and positive semidefinite). This formulation has the disadvantage that the predicted sampling probabilities depend on the covariance matrix determined by $A_{sample}$ rather than $A$, but this covariance matrix may be natural in certain contexts.

Furthermore, while the diagonal terms sampled with probability $p_{ii}$ do not correspond to edges, we can still fix them with no harm to our theoretical results (i.e., set $p_{ii} = 1$ as noted earlier in the oversampling paragraph of this section).

**Synapses with probabilities greater than** 1  A calculated probability term for a synapse $p_{ij}$ that is $> 1$ can be handled in two ways. One solution is to convert each synaptic weight into pieces with predicted probability $< 1$ and rewrite the sum in Eq. 5 as involving multiple pieces corresponding to the edge $(i, j)$ each sampled with probability 1. Note that this will increase $K_{deg}$ slightly but does not change the actual form of the sampling rule (the edge is just preserved). A second approach is to split the matrix into a deterministic and a sampled piece, and apply the argument to the sampled piece (as in the argument for fixed edges above). Again this has the drawback that the predicted sampling probabilities would not be given by the covariance matrix of the entire network.

# S2    Extensions

## S2.1    Near-diagonally-dominant networks

Let the matrix $A$ be a (not necessarily diagonally-dominant) symmetric negative definite matrix corresponding to the coupling matrix of a linear system such as in Eq. 1. We analyze the effect of applying noise-prune to $A$ in terms of its distance from a diagonally dominant matrix.

As before, we let $B = -A$ and analyze the effect of the rule on $B$. Note that the noise-driven covariance matrix of the linear system $C \propto -A^{-1} = B^{-1}$, and that the sampling probabilities yielded by noise-prune are $p_{ij} = K|w_{ij}|(C_{ii} + C_{jj} - 2\operatorname{sgn}(w_{ij})C_{ij}) = K_{deg}|w_{ij}|(B_{ii}^{-1} + B_{jj}^{-1} - 2\operatorname{sgn}(w_{ij})B_{ij}^{-1})$ for $i > j$. We will also set any excess diagonal probabilities $p_{ii} = 1$ (note that this is implicitly done in both the original and matched diagonal settings of noise-prune in the main paper).

Set $\gamma > 0$ and define the matrix $B_\gamma = B + \gamma I$. Let $B$ have eigenvalues $\lambda_k$ and eigenvectors $\boldsymbol{v}_k$ and observe that $B_\gamma$ has eigenvalues $\lambda_i + \gamma$ and the same eigenvectors as $B$. Moreover, note that applying noise-prune to $B$ with some set of probabilities to yield $B^{sparse}$ is equivalent to applying noise-prune to $B_\gamma$ with the same set of probabilities to yield $B_\gamma^{sparse} = B^{sparse} + \gamma I$ (though these are not the optimal probabilities for $B_\gamma$).

Now take $\gamma$ large enough so that $B_\gamma$ is diagonally dominant (the approximation described below will be good if $\gamma$ is small). The framework described in Section S1.1 can then be applied to $B_\gamma$ and the probabilities saturating the Chernoff bound are $p_{ij}^{(\gamma)} = K_{deg}|w_{ij}|([B_\gamma]_{ii}^{-1} + [B_\gamma]_{jj}^{-1} - 2\operatorname{sgn}(w_{ij})[B_\gamma]_{ij}^{-1})$.

In particular, note that

$$[B_\gamma]_{ij}^{-1} = \left(\sum_k \frac{1}{\lambda_k + \gamma}\boldsymbol{v}_k\boldsymbol{v}_k^T\right)_{ij} = \sum_k \frac{1}{\lambda_k + \gamma}\left(\boldsymbol{v}_k\boldsymbol{v}_k^T\right)_{ij} = \sum_k \frac{1}{\lambda_k + \gamma}(\boldsymbol{v}_k)_i(\boldsymbol{v}_k)_j \quad \forall i,j \quad (33)$$

and similarly

$$B_{ij}^{-1} = \left(\sum_k \frac{1}{\lambda_k}\boldsymbol{v}_k\boldsymbol{v}_k^T\right)_{ij} = \sum_k \frac{1}{\lambda_k}\left(\boldsymbol{v}_k\boldsymbol{v}_k^T\right)_{ij} = \sum_k \frac{1}{\lambda_k}(\boldsymbol{v}_k)_i(\boldsymbol{v}_k)_j \quad \forall i,j, \quad\quad (34)$$

where $(\boldsymbol{v}_k)_\ell$ denotes the $\ell$th entry of the $k$th eigenvector $\boldsymbol{v}_k$. Then we can see that, for $i > j$,

$$p_{ij}^{(\gamma)} = K_{deg}|w_{ij}| \sum_k \frac{1}{\lambda_k + \gamma}\left((\boldsymbol{v}_k)_i^2 + (\boldsymbol{v}_k)_j^2 - 2\operatorname{sgn}(w_{ij})(\boldsymbol{v}_k)_i(\boldsymbol{v}_k)_j\right)$$

$$= K_{deg}|w_{ij}| \sum_k \frac{1}{\lambda_k + \gamma}((\boldsymbol{v}_k)_i - \operatorname{sgn}(w_{ij})(\boldsymbol{v}_k)_j)^2$$

$$\leq K_{deg}|w_{ij}| \sum_k \frac{1}{\lambda_k}((\boldsymbol{v}_k)_i - \operatorname{sgn}(w_{ij})(\boldsymbol{v}_k)_j)^2$$

$$= p_{ij},$$

where the inequality follows from the fact that $\frac{1}{\lambda_k + \gamma} \leq \frac{1}{\lambda_k}$ and the rest of the terms in the expression are all nonnegative.

Thus $p_{ij}^{(\gamma)} \leq p_{ij}$ for all $i \geq j$ and sparsifying $B_\gamma$ using the probabilities $p_{ij}$ yields Eq. 7 for at most the same degree of error $\epsilon$ we would get if we used the $p_{ij}^{(\gamma)}$'s instead (see Section S1.4 for more details on oversampling). That is,

$$(1 - \epsilon)\boldsymbol{x}^T B_\gamma \boldsymbol{x} \leq \boldsymbol{x}^T (B_\gamma)^{sparse}\boldsymbol{x} \leq (1 + \epsilon)\boldsymbol{x}^T B_\gamma \boldsymbol{x} \quad \forall \boldsymbol{x} \in \mathbb{R}^N. \tag{35}$$

Now observing that $(B_\gamma)^{sparse} = B^{sparse} + \gamma I$ allows us to subtract $\boldsymbol{x}^T \gamma I \boldsymbol{x}$ through our inequality to arrive at

$$(1 - \epsilon)x^T B\boldsymbol{x} - \epsilon\gamma\boldsymbol{x}^T \boldsymbol{x} \leq \boldsymbol{x}^T B^{sparse}\boldsymbol{x} \leq (1 + \epsilon)\boldsymbol{x}^T B\boldsymbol{x} + \epsilon\gamma\boldsymbol{x}^T \boldsymbol{x} \quad \forall \boldsymbol{x} \in \mathbb{R}^N. \tag{36}$$

In other words, sparsifying $B$ using the same probabilities we used for $B_\gamma$ guarantees a result similar to that of Eq. 7, but with an additional additive error of $\epsilon\gamma\boldsymbol{x}^T \boldsymbol{x}$. In particular, if $\boldsymbol{x}$ has unit norm then the additive error is simply $\epsilon\gamma$.

## S2.2  Rectified linear units

Define the rectified linear activation function $[\cdot]_+ = \max[0, \cdot]$ and consider the recurrent neural network

$$\frac{d\boldsymbol{x}}{dt} = -D\boldsymbol{x} + [W\boldsymbol{x} + \boldsymbol{b}(t)]_+. \tag{37}$$

As before, define $A = -D + W$, and let $A^{sparse}$ be the result of applying noise-prune to $A$ using the probabilities from the linear network defined by $A$ (so that Eq. 7 holds for $A$ and $A^{sparse}$).

Let $\Gamma(t) = \{i : \sum_j W_{ij}x_j + b_j(t) > 0\}$ be the indices of neurons that receive suprathreshold input at time $t$. Define $A_{\Gamma(t)}$ and $A_{\Gamma(t)}^{sparse}$ to be the submatrices produced by removing the rows and columns of $A$ and $A^{sparse}$ corresponding to indices not in $\Gamma(t)$. We will show that the dynamics of the network in Eq. 37 are approximately determined by the set of linear systems (indexed by $t$) with coupling matrices $A_{\Gamma(t)}, A_{\Gamma(t)}^{sparse}$. In other words, the dynamics of a rectified linear network switch among the dynamics of a set of linear networks, with the appropriate linear network at a moment in time determined by the subset of neurons that receive suprathreshold input (see [9, 10] for more on this argument).

For convenience, let $\Gamma(t)^c$ be the complement of $\Gamma(t)$; that is, $\Gamma(t)^c$ is the collection of neurons that receive zero input. The neurons in $\Gamma(t)^c$ either have zero activity (and thus can be ignored) or have nonzero activity but receive zero input (and thus contribute feedforward input to the rest of the network that can be absorbed into the input vector). Define $\boldsymbol{x}_{\Gamma(t)}$ and $\boldsymbol{b}_{\Gamma(t)}$ to be the vectors produced by removing the entries of $\boldsymbol{x}$ and $\boldsymbol{b}$ corresponding to the indices in $\Gamma(t)^c$, as well as $\boldsymbol{x}_{\Gamma(t)^c}$ to be the vector produced by removing the entries of $\boldsymbol{x}$ corresponding to the indices in $\Gamma(t)$. Lastly, define $\delta\boldsymbol{b}_{\Gamma(t)}$ to be the feedforward contribution of $\boldsymbol{x}_{\Gamma(t)^c}$ (more precisely, the $i$th entry of this vector is given by $\sum_{j\in\Gamma(t)^c} W_{ij}x_j$ with $i \in \Gamma(t)$ listed in increasing order) that we will absorb into the new input vector for our smaller system in $\mathbb{R}^{|\Gamma(t)|}$, defined to be $\tilde{\boldsymbol{b}}_{\Gamma(t)} = \boldsymbol{b}_{\Gamma(t)} + \delta\boldsymbol{b}_{\Gamma(t)}$.

Now the dynamics of the network in some small time interval around $t$ are determined by the linear system,

$$\frac{d\boldsymbol{x}_{\Gamma(t)}}{dt} = A_{\Gamma(t)}\boldsymbol{x}_{\Gamma(t)} + \tilde{\boldsymbol{b}}_{\Gamma(t)}. \tag{38}$$

And the nodes in $\Gamma(t)^c$ either have 0 activity or are decaying to 0 with the leak time constant.

Note that Eq. 7 holds for $A_{\Gamma(t)}, A_{\Gamma(t)}^{sparse}$ as well. Let $\Gamma(t,j)$ be the index of the $j$-th active neuron at time $t$. Given $\boldsymbol{x}_{\Gamma(t)} \in \mathbb{R}^{|\Gamma(t)|}$, consider the natural extension vector $\boldsymbol{x} \in \mathbb{R}^N$ whose entry in $\Gamma(t,j)$ is the $j$ entry of $\boldsymbol{x}_{\Gamma(t)}$ and whose entries in $\Gamma(t)^c$ are 0. Then $\boldsymbol{x}_{\Gamma(t)}^T A_{\Gamma(t)}^{sparse} \boldsymbol{x}_{\Gamma(t)} = \boldsymbol{x}^T A^{sparse} \boldsymbol{x}$ (and similarly, $\boldsymbol{x}_{\Gamma(t)}^T A_{\Gamma(t)} \boldsymbol{x}_{\Gamma(t)} = \boldsymbol{x}^T A \boldsymbol{x}$), so the fact that Eq. 7 holds for $A, A^{sparse}$ implies that it holds for $A_{\Gamma(t)}, A_{\Gamma(t)}^{sparse}$ (for all $t$). Thus, among other quantities, the spectrum of $A_{\Gamma(t)}$ is approximately preserved (to within $\epsilon$) by $A_{\Gamma(t)}^{sparse}$. Thus, we see that noise-prune preserves the dynamics of linear systems described the submatrices $A_{\Gamma(t)}$. Finally, $\tilde{\boldsymbol{b}}_{\Gamma(t)}$ depends on the weights through $\delta\boldsymbol{b}_{\Gamma(t)}$, which may be perturbed in the sparse system, though it is preserved in expectation. However, perturbations are likely to be small because this additional feedforward input comes from the small subset of low-activity neurons in $\boldsymbol{x}_{\Gamma(t)^c}$ that receive sub-threshold input and are approaching zero activity but have not completely decayed yet (which they do so with time-constant given by the leak). In short, the dynamics of a rectified linear network are approximately preserved when its coupling matrix is sparsified in the same manner as that of a linear network.