[Reviews · NeurIPS 2020]

Review 1

Summary and Contributions: In this paper, the authors describe a novel approach to prune the synapses (the weights) in a neural network. More in detail, the proposed method prunes or strengthen the weights only analyzing the local information (i.e., the covariance of the neighbor neurons). This local plasticity rule, inspired by the biology, is completely unsupervised. It tries to preserve the network dynamics, while reducing the number of useless connections. The correctness of the proposed solution has been proven under some constraints on the shape of the network matrix.

Strengths: The main strength of the proposed algorithm is its completely local, unsupervised and biologically plausible approach that can work in parallel with any other supervised and unsupervised training algorithm. It is not affected by the network structure and it can be also applied to recurrent networks. It only uses local information and it’s simple. Finally the correctness is proven under some constrains and the authors state that in this regime, where theoretical conditions are satisfied, the algorithm works pretty fine, drastically sparsifying the edges while preserving the network properties.

Weaknesses: The main weakness of the proposed algorithm is its limited applicability. Indeed, it can be only applied to linear neural networks, eventually rectified. The proof of correctness is conditioned by two strong limitations. The network matrix is required to be symmetric and diagonally dominant. This is not a typical network as it is an indirect graph with strong auto-recursion for each neuron. Finally experimental validation is missing. An extensive evaluation with a real problem, including other pruning approaches, would help to better understand whether the guiding principle works in the right direction.

Correctness: The theory outlined in this work is solid and apparently correct. The claimed outcomes of the presented approach seem reasonable and are supported by the theory. The empirical validation, on the contrary is missing.

Clarity: The paper is well written. Figures are a bit too small.

Relation to Prior Work: The “Discussion” section contains some comments on how the presented approach differs from a category of existing methods pruning the weights. However neural networks pruning has a long history. I think a slightly more detailed discussion of the state of the art would improve the paper. As already mentioned, the section “Experiments” is nearly missing. For the sake of completeness, the paper would need a comparison with state of the art approaches.

Reproducibility: Yes

Additional Feedback: Post rebuttal: I found convincing the answer regarding the applicability to generic networks and I adapt the overall score accordingly, nevertheless, it is not that fair to add a new part to the paper after the review.


Review 2

Summary and Contributions: The authors of the paper "Using noise to probe recurrent neural network structure and prune synapse" present a novel scheme to sparsify linear dynamical systems (linear recurrent neural networks) and show that it is extendable to rectified linear recurrent neural networks. This rule consists just of local terms, and requires a phase where noise is injected in the network, hence rendering it plausible in terms of brain computation. Also the rule is meant to preserve network dynamics, therefore has no intention to preserve performance in specific tasks (unsupervised). The main contributions are the pruning rule "noise-prune" and a solid theoretical analysis/derivation that supports the proposed rule.

Strengths: The main strengths of the work are a seemingly simple pruning rule, which is derived from first principles using results from graph sparsification. The work has a strong theoretical grounding, where the single pieces combined end up in a simple form. The analysis and derivation provide a significant contribution and I believe it is relevant to NeurIPS community because sparse connectivity is interesting from the perspective of both Neuroscience and Machine Learning.

Weaknesses: One weakness is that the theory applies only to symmetric connectivity matrices and linear systems (with one exception), but I do not view this as a major weakness given the theoretical results of the paper.

Correctness: The derivation and methodology appears correct to me, yet I could not check it into the finest details.

Clarity: The present work reads nicely and thoroughly discusses the obtained results. There is one exception though, one could make the derivation steps around Eq. 8 clearer, i.e. it would be very helpful for a reader to provide more detail here similar to what was done in the supplement. In addition, I find it confusing that the authors refer in l.131 to Fig. 1c, where the diagonal terms on the right hand side are stars but one of them would always be zero in each matrix.

Relation to Prior Work: Prior work seems to be adequately addressed.

Reproducibility: Yes

Additional Feedback: I believe Fig. 1c could be improved. At least by showing a zero instead of a * star in at the diagonal elements which actually are zero on the right hand side (1 out of 3 diagonal elements). I also believe that Fig. 2 could be substantially improved. I would recommend here to present the samples here as proper violin plots which are much wider than they are currently. It would facilitate reading the Figure. The ticks could also be more sharp. Post rebuttal update: My score remains unchanged and I hope that the quality of the Figures will be improved.


Review 3

Summary and Contributions: This paper proposed a potential use of noise in the nervous system—for pruning synapses—which is an innovative idea. The authors developed a simple, local, unsupervised plasticity rule that relies only on synaptic weights and noise-driven covariance of neighboring neurons. For a subset of linear and rectified linear networks, they prove this rule preserves the spectrum of the original matrix and preserves network dynamics even for high levels of sparsification. This is an interesting perspective and plausible as a biological mechanism, though by no means proven. It suggests a new role for neuronal variability in neural computation, which is rather orthogonal to the sampling hypothesis that is popular nowadays, and thus a breath of fresh air.

Strengths: Synaptic pruning in neural networks typically is based on the strength of the connection weights. This paper championed the novel idea that noise correlation can be used to set up criteria to attenuate or prune redundant connections and provided reasonable theoretical proofs for specific types of networks. The authors are honest about the limitations. The framework may apply more naturally to subnetworks with a high probability of reciprocal connectivity.

Weaknesses: The demonstration examples provided could be more compelling. Specifically, simulations of a network with known/interesting low-dimensional dynamics, and results that those dynamics are preserved after pruning would be ideal. The proof was limited to symmetric diagonal dominant linear and rectified linear neurons. Empirical results /demonstrations are a bit underwhelming.

Correctness: Reasonable approach. I believe they are correct.

Clarity: Well written.

Relation to Prior Work: There are a lot of works on network pruning, but the idea of using noise correlation is innovative and interesting.

Reproducibility: Yes

Additional Feedback: I provided these comments in the Discussion to argue for its acceptance: I found the authors' responses addressed the issues of "symmetric connections" and "biological plausibility" reasonably well. Both the reviewers who gave "5" agreed that the theoretical derivation is correct. They mostly questioned the biological plausibility or applicability. While symmetric connections are not necessarily biological plausible, many important models and theoretical analysis, for example the works of Hopfield, Sompolinsky etc, often made such simplifying assumptions and produced works that have been influential in theoretical neuroscience at the end. I liked the paper because the idea is interesting, novel and innovative, that the learning and pruning can be local using noise as probe has not been proposed or explored before. It gave us a new perspective on noise correlation. It is very plausible that this might turn out not to be true in the real biological reality, but isn't that true for most models as well. I don't see that as a major problem, and I feel this idea deserved to be considered seriously, examined and tested. Furthermore, the authors' rebuttal claimed that they will include positive empirical results on networks with non-symmetric connections. The authors' response to Reviewer 4's philosophical comments is also reasonable to me. Pruning might serve many functional purposes and might need to satisfy multiple constraints. As I said originally , this is not a perfect paper, but I prefer to see it as half full. Among the bunch of papers I reviewed this year, this is the only one I found refreshing and interesting. "8" is my highest score.


Review 4

Summary and Contributions: The authors propose a synaptic pruning algorithm based on stimulating a neural network with noise and measuring covariances. They prove that such pruning algorithm preserves the dynamical properties of the connectivity matrix.

Strengths: This is a theoretically interesting contribution.

Weaknesses: I am skeptical about the biological applicability of the results. However, based on the author feedback and reviewer discussion I raise my score.

Correctness: Yes

Clarity: Yes

Relation to Prior Work: Yes

Reproducibility: Yes

Additional Feedback: While it is likely that the sparsification of neural networks in the brain helps conserve limited resources such as space and metabolic energy, I suspect that synaptic pruning in the brain plays a computational role not captured solely by preserving the dynamical properties of the network in the authors' model. Therefore, I am skeptical of the biological applicability of the authors' model. For example, consider the following trade off thought to arise due to synapse elimination. Overproduction of synapses early in development endows the brain with a huge plasticity potential (or flexibility) allowing the neural network to easily adapt to a great variety of possible environmental conditions it may face (e.g. the ability to learn foreign languages). As synapses get pruned such flexibility is lost (e.g. try learning a foreign language in the middle age). Yet, synapse elimination is beneficial because the remaining synapses typically get stronger, making the network dynamics more precise, fast, efficient and reproducible (e.g. consider language skills developing with age).

[Author Response · NeurIPS 2020]

We thank the reviewers for their thoughtful comments and will make the suggested changes to clarify figures and steps of the derivation.

**Expanded empirical validation** We will use most of the extra page in the revised version to expand the section on empirical results and in our plots will now show network dynamics as well as spectra.

We have empirically characterized noise-prune's performance on non-symmetric clustered networks (i.e., going beyond the theory) and find it works very well. The figure included below shows sample plots, comparing noise-prune to a control that prunes only based on weight. These empirical results thus extend beyond the theory and show that the guiding principle works in the right direction. We will include an expanded set of these results in the manuscript.

The current state of the art in pruning in machine learning prunes while preserving a task-dependent cost function and either uses multiple steps of training and pruning or complex non-local measures of synaptic importance. We will add a more detailed discussion of these algorithms. We have not yet characterized noise-prune's performance against these algorithms (we intend to in future work). But we would be surprised if noise-prune performed better given that it is so simple, uses only local information, and seeks to preserve all dynamics rather than performance on a specific task. To avoid any confusion, we will explicitly state that we do not expect noise-prune to perform better than these algorithms.

We believe our work is nevertheless of broad interest to researchers studying many types of neural networks and not just to researchers interested in unsupervised learning, dynamical systems and computational neuroscience. As the reviewers point out, we provide a novel perspective and strong theoretical results on the problem of pruning. These may contribute to future algorithms. Moreover, we provide a bridge between pruning and graph sparsification, noise-driven dynamical systems, and matrix concentration of measure techniques, which are all active areas of research with many more possibly fertile ideas for pruning.

We plan to follow this study (focused on introducing and developing the theory) with an extensive empirical characterization of noise-prune's performance on nonlinear and non-symmetric networks, but believe (and hope that the reviewers agree) that the inclusion of some numerical results on non-symmetric networks (in addition to expanded results on symmetric networks) in the current study will adequately support and point beyond the theory.

**Biological applicability (R4):** We agree that the computational role of synaptic pruning is likely not captured solely by preservation of dynamics. However, we would argue that a good synaptic pruning rule should at least preserve the broad pattern of dynamics. Building off of the Reviewer's language learning example, even here the dynamical patterns are likely to be similar between unpruned and pruned networks, because they need to carry out a similar set of input-output transformations, even if the pruned network is faster and more reliable. Thus dynamics preservation could be used as a building block for more complex pruning algorithms. For example, note in the model the non-pruned synapses are also strengthened. If the model included multiplicative fluctuations in synapse strengths (as suggested by spine head size fluctuations), then non-pruned strengthened synapses would be more reliable, making dynamics more precise and reproducible. Or, adding nonlinearities to downstream neurons could make these synapses faster or more efficient at driving downstream activity. We will now include some text on other benefits of sparsity in the Discussion.

Figure 1: **Noise-prune on non-symmetric clustered networks**. Networks have dense within-cluster and sparse between-cluster connections. Black traces are original unpruned network; red traces are networks pruned to 20% sparsity with noise-prune; blue traces are pruned to 20% based only on weights. (a) Response to a random input at time 0. Left: response of 4 different network nodes, showing activity over time. Noise-prune (red) and the original network (black) are very close, with black sometimes covered by red. Right: Difference between network activity vectors before and after pruning. Red trace shows $||x_{orig}(t) - x_{np}(t)||_2/||x_{orig}(t)||_2$, where $x_{orig}(t)$ is the original network's activity vector in response to a random input and $x_{np}$ is the equivalent for the noise-prune network. Blue shows equivalent curve for weight-based pruning. (b) Results in (a) were for a random input. This panel shows results for an input along one of the slow eigenmodes of the network. Such slow eigenmodes are thought to be important for maintaining information over time in biological networks. Noise-prune performs significantly better than pruning by weights.

[Meta-Review · NeurIPS 2020]

All the reviewers agree that the theoretical derivation of the model is correct. The authors satisfactorily replied in the rebuttal to the main concern on the symmetric connections. The contribution of this manuscript is considered innovative and of interest even though the biological plausibility might turn out not be true.